# Patent Foramen Ovale and Cryptogenic Stroke: Integrated Management

**DOI:** 10.3390/jcm12051952

**Published:** 2023-03-01

**Authors:** Fabiana Lucà, Paolo G. Pino, Iris Parrini, Stefania Angela Di Fusco, Roberto Ceravolo, Andrea Madeo, Angelo Leone, Mark La Mair, Francesco Antonio Benedetto, Carmine Riccio, Fabrizio Oliva, Furio Colivicchi, Michele Massimo Gulizia, Sandro Gelsomino

**Affiliations:** 1Cardiology Department, Grande Ospedale Metropolitano, GOM, AO Bianchi Melacrino Morelli, 89129 Reggio di Calabria, Italy; 2Cardiology Department, San Camillo Forlanini Hospital, 00152 Rome, Italy; 3Cardiology Department, Mauriziano Hospital, 10128 Torino, Italy; 4Clinical and Rehabilitation Cardiology Department, San Filippo Neri Hospital, ASL Roma 1, 00135 Roma, Italy; 5Cardiology Unit, Giovanni Paolo II Hospital, 88046 Lamezia, Italy; 6Ferrari Hospital, 87012 Castrovillari, Italy; 7Ospedale Santissima Annunziata, 87100 Cosenza, Italy; 8Cardiothoracic Department, Brussels University Hospital, 1090 Jette, Belgium; 9Division of Clinical Cardiology, A.O.R.N. ‘Sant’Anna e San Sebastiano’, 81100 Caserta, Italy; 10De Gasperis Cardio Center, Niguarda Hospital, 20162 Milano, Italy; 11Cardiothoracic Department, Garibaldi Nesima Hospital, 95122 Catania, Italy; 12Cardiothoracic Department, Maastricht University, 6211 LK Maastrich, The Netherlands

**Keywords:** Patent foramen ovale (PFO), device closure, interatrial septal aneurysms, PFO-associated syndromes, cryptogenic stroke

## Abstract

Patent foramen ovale (PFO) is a common cardiac abnormality with a prevalence of 25% in the general population. PFO has been associated with the paradoxical embolism causing cryptogenic stroke and systemic embolization. Results from clinical trials, meta-analyses, and position papers support percutaneous PFO device closure (PPFOC), especially if interatrial septal aneurysms coexist and in the presence of large shunts in young patients. Remarkably, accurately evaluating patients to refer to the closure strategy is extremely important. However, the selection of patients for PFO closure is still not so clear. The aim of this review is to update and clarify which patients should be considered for closure treatment.

## 1. Introduction

Patent foramen ovale (PFO) is a congenital cardiac abnormality, usually incidentally found [1]. An incidence of 20–30% has been reported in the general population [1,2]. Remarkably, it can be associated with other pathologic conditions such as cryptogenic stroke (CS), decompression illness (DCI) [3], secondary migraine headache (MHA) [4], arterial deoxygenation and platypnea-orthodeoxia [1,5,6], which have been defined as PFO-associated syndromes. It is worth mentioning that one-third of ischemic strokes are considered to be cryptogenic, and the majority of them have been thought to be caused by paradoxical embolism due to PFO [7,8].

In the early trials, the superiority of interventional strategy versus medical therapy has not been confirmed. However, more recently it has been suggested that the percutaneous PFO closure (PPFOC) with the latest devices plays a significant role in reducing clinical events [9,10,11,12,13,14]. According to the latest international guidelines, PFO closure and long-term antiplatelet therapy over antithrombotic therapy alone should be recommended in patients aged between 18 and 60 years with a previous stroke or transient ischemic attack (TIA) [14,15,16]. 

However, selecting who can really benefit from the interventional approach is extremely important.

This review aims to better clarify which patients should be considered for closure treatment according to a multiparametric approach.

## 2. Epidemiology and Classification

A cryptogenic mechanism seems to be involved in 10–40% of all ischemic strokes [17,18]. 

PFO is thought to cause about 50% of cryptogenic strokes [8,19]. Nevertheless, other causes can be involved in the pathogenesis of the ischemic event, such as subclinical atherosclerosis, unstable plaques at intracranial and carotid vessels, arterial dissection, vasculitis, hypercoagulable states, other cardioembolic sources and atrial fibrillation (AF) [20].

Ischemic stroke can be classified by the TOAST classification [21], the ASCOD phenotyping system (A: atherosclerosis; S: small-vessel disease; C: cardiac pathology; O: other cause; D: dissection) [22] and the Causative Classification System [23]. If the full diagnostic work-up results in excluding other etiologies, stroke can be considered cryptogenic [24]. Additionally, CS can be categorized as non-embolic or embolic [25]. The embolic stroke of undetermined source (ESUS) definition involves non-lacunar brain infarcts without proximal arterial stenosis or major cardioembolic sources [24]. Not only is paradoxical thromboembolism linked to PFO, but also to AF, hypercoagulability, antiphospholipid syndrome, cancer and minor embolic sources. Pulmonary fistula has also been shown to underline CS [25]. As opposed to events caused by AF, PFO-related strokes are likely to radiologically manifest as a single cortical lesion or as numerous small localized injuries, mainly placed in the vertebrobasilar areas [26]. Classifications of causative stroke mechanisms, general nomenclature, and classification for PFO-associated stroke have recently been updated [27]. An accurate revision of the main stroke subtyping algorithms has also been provided. The term “PFO-associated stroke” has been proposed to indicate a distinct causative mechanism of ischemic stroke, which may be used after a comprehensive patient evaluation [27]. If superficial, large, deep, or retinal infarcts occur in patients with a medium- to high-risk PFO without other demonstrated causative factors, we can name the event a PFO-associated stroke [27]. 

## 3. Imaging Assessment of Suspected Paradoxical Embolism

Echocardiography plays an essential role in detecting anatomical and functional aspects of PFO. Transthoracic echocardiography (TTE) represents the first diagnostic step for detecting PFO [28]. Notably, its sensitivity is low (46%); however, it significantly increases if harmonic imaging is performed [29,30,31]. Moreover, if agitated saline contrast is used [32], its accuracy in identifying the paradoxical right to left shunts due to PFO and assessing residual shunts after PPFOC significantly improves [33,34].

Provocative maneuvers are useful in demonstrating transient undetected paradoxical shunts [1]. 

It has been well established that transesophageal echocardiography (TEE) is accurate in detecting identifying PFO, providing an assessment of shunt dimension, characterizing the anatomic features and allowing differential diagnosis from other conditions such as atrial septal defect [35], as well as thromboembolic sources (complex aortic plaques, fibroelastoma, or valvular vegetations) (Figure 1).

Transcranial Doppler (TCD) with bubble study has been shown to be effective in detecting the presence of right-to-left shunting and, consequently, in indirectly demonstrating PFO [36]. Remarkably, if TCD is negative, no further studies are required. 

The Spencer logarithmic scale has been used for assessing the severity of shunting [37], classifying five grades (0–5) from the absence of a shunt (0) to a severe shunt (5). A grade ≥ 3, according to the Spencer scale, is considered diagnostic [37]. Nevertheless, although it has a higher sensitivity (97%), it is limited in distinguishing cardiac from pulmonary shunting, resulting in a lower specificity (93%) [36].

An initial evaluation with transcranial Doppler, followed by performing TEE if needed, is a widespread approach [38,39]. Indeed, the positivity of this test does not allow us to accurately identify whether the shunting originates from PFO or atrial septal defect, or if it is intrapulmonary.

A three-dimensional echocardiography (3DE) TEE approach provides a more comprehensive morphologic assessment compared to TTE; in addition, it can provide incremental information that is helpful in choosing between percutaneous transcatheter and surgical closure [40], such as the size of PFO and precise location of shunting. Finally, combined modalities imaging can be used to obtain a more accurate detection of PFO [33] (Figure 1 and Figure 2).

Angiography is not indicated to diagnose PFO, except for the cases in which it is performed for other reasons [41].

Cardiac MRI is less frequently used for the detection of PFO because of its low sensitivity.

## 4. Thrombophilia Screening in Cryptogenic Stroke

Screening for coagulation disorders should also be considered in specific cases. Indeed, although this screening has an overall low diagnostic yield, in subgroups of younger patients in particular, such as those without vascular risk factors, those with recurrent venous or arterial thrombotic events, and those with a family history of thrombosis, the diagnostic yield is higher. Overall, the reported prevalence of hypercoagulable states in ischemic stroke is 3–21% [42]. Currently, the evidence does not support thrombophilia screening as a routine practice in ischemic stroke [43]. In addition, the results of thrombophilia tests seem to have a low likelihood of impacting outpatient management [44]. Among the inherited coagulation disorders, prothrombin gene mutation and deficiencies of protein C, protein S, and antithrombin III seem to have a weak association with the risk of ischemic stroke [45]. More evidence is available on the possible clinical relevance of factor V mutation [46] and the MTHFR mutation [47] in ischemic stroke. However, even for these coagulation disorders, the current evidence is not strong.

Of note, some coagulation tests, such as protein S and protein C activity, may be falsely abnormal in the acute setting of a thrombotic event. Therefore, they should be performed some weeks after the critical event when clinically required. Furthermore, considering the substantial cost of thrombophilia testing, and to avoid performing tests without a clinical impact (which does not impact the following clinical management), a specialist should be consulted before ordering testing for inheritable thrombophilia [44]. Testing results must be interpreted by an experienced clinician, considering all factors that may influence test results in every case. According to the American guidelines for stroke prevention, anticoagulant treatment may be considered in patients with inherited thrombophilic states. If anticoagulant therapy (OAC) is not employed, antiplatelet therapy (APT) is recommended [45]. In young patients, acquired hypercoagulable states—especially the persistent presence of antiphospholipid antibodies—seem to be a leading cause of ischemic stroke [48]. Therefore, screening for this condition is warranted in younger patients with cryptogenic stroke. In patients with stroke and antiphospholipid antibodies, but who do not fulfill the criteria for antiphospholipid syndrome, antiplatelet treatment is recommended [45]. However, in patients with stroke and a diagnosis of antiphospholipid syndrome, OAC might be considered when also taking into account the overall thrombotic and bleeding risk in every single case [45]. In patients with antiphospholipid syndrome, if the OAC is not employed, APT is recommended [45].

## 5. Medical and Interventional Treatment

Establishing the etiological role of PFO as the cause of stroke, and excluding other causes of stroke is essential in choosing the strategy to follow.

Currently, three different therapeutic options have been proposed for the secondary prevention of stroke in PFO patients: APT, oral OAC, and PPFOC [14].

Although a significant benefit of PPFOC versus medical therapy for the secondary prevention of cryptogenic stroke has not been confirmed in previous studies [49,50,51], a more recent statistically significant benefit in terms of the composite of stroke, TIA, and death reduction has been reported in patients referred to PPFOC [12].

The superiority of PPFOC versus medical treatment in the prevention of stroke, particularly for patients with shunts of considerable size has been shown [52], although controversial data exist [53]. However, the recurrence of stroke has been estimated to be low in both in the interventional and medical strategy, consisting of an annual risk of 0.61% and 1.17%, respectively. Remarkably, PPFOC significantly increased the risk of AF [54,55,56]. Female sex and a greater size of PFO (>30 mm) are considered to be independent predictors of complication. Conversely, a beneficial effect of PPFOC has also been reported in male and young patients (≤45 years old) [57].

However, procedural-related events such as aortic erosion, device thrombosis, embolization, endocarditis and atrio-aortic fistula are quite uncommon.

The CLOSE trial [13] enrolled 663 patients with sizeable interatrial shunt or atrial septal aneurysms with a previous PFO-related stroke. Patients were randomized 1:1:1 to PPFOC, plus APT vs. APT alone vs. OAC. The CLOSE trial confirmed the long-term follow-up data of RESPECT [9], demonstrating that PPFOC significantly reduced recurrent stroke in patients with CS who have specific echocardiographic features such as a septal aneurysm or a large interatrial shunt. Remarkably, oral OAC was associated with a non-significant stroke risk reduction compared to APT.

Nevertheless, the NAVIGATE-ESUS trial, investigating aspirin vs. rivaroxaban for the secondary prevention of stroke and systemic embolism, was prematurely stopped because there was no difference in efficacy between the two arms and an excess of bleeding with rivaroxaban [58].

In the GORE REDUCE trial, [13] 664 enrolled patients with cryptogenic stroke and PFO were randomized 2:1 to PPFOC, plus APT or APT alone. Results confirmed the meta-analysis of earlier RCTs and long-term data from RESPECT and CLOSE, where PPFOC was superior to medical treatment in patients with cryptogenic stroke.

A recent meta-analysis [52] from the five RCTs pointed out that two-thirds can reduce the risk of recurrent stroke by PPFOC in addition to medical therapy, with an increase of benefit depending on the shunt size. Atrial fibrillation was rare, but significantly increased in the closure arm vs. medical treatment. A stroke is known to lead to potentially dramatic consequences, and several patients consider it a worse outcome than death [59]. A pro-inflammatory mechanism related to the PPFOC procedure has been recognized to underline AF. The need for anti-arrhythmic prophylaxis, OAC, and its duration remains debated in clinical practice [60].

A meta-analysis of 48 observational studies (10,327 patients) demonstrated that in patients with cryptogenic stroke or TIA, PPFOC was associated with a reduction of 6.25 fold in neurological events compared with medical therapy [61].

Regarding the safety of PPFOC, the data from clinical trials and meta-analysis showed an incidence of procedural-related events of 3.2%, with significant differences in all-cause major adverse events. Chest pain is an occasional complication associated with the device, and is considered secondary to an excess of inflammatory response and scar tissue formation, likely due to nickel allergy [62].

A recent meta-analysis examining the difference between PPFOC and medical therapy in terms of stroke recurrence [63] has evaluated whether the treatment response was affected by the Risk of Paradoxical Embolism (RoPE) score [19] and PFO-Associated Stroke Causal Likelihood (PASCAL) Classification System [27]. According to the RoPE score evaluation (Table 1), aimed at investigating the likelihood that a PFO may cause a PFO-related stroke, larger values have been associated with a higher probability [19]. Indeed, due to the high prevalence of PFO in the general population (roughly 25%) [64], whether stroke is caused by a PFO-related mechanism or other underlying factors may be difficult to determine. Remarkably, the RoPE score was limited by a lack of comprehensive patient-selection-related information and thus integrated by the PASCAL Classification System (Table 2). Evaluating patients according to the RoPE score and the PASCAL Classification could be helpful in identifying those patients who might benefit most from PPFO [65,66,67,68]. In patients younger than 60 years without a detected cause of stroke, the indication for PPFOC is suggested by a RoPE score > 7 and/or a large shunt or interatrial septal aneurysm associated with a probable or certain probability that the stroke is causally related to the PFO, according to the PASCAL Classification System (Table 2).

In patients over 60 years of age with coronary risk factors and an increased association with atherosclerotic disease, the question of PPFOC remains open, considering the fact that elderly patients have been substantially excluded from the principal trials, except for a study that evaluated older recipients [69]. As with the younger population, the prevalence of PFO has been shown to be higher in elderly patients with CS than in peer patients with a known cause of ischemic stroke [70]. Thus, CS might be caused by PFO in the elderly as well [71]. Early and long-term clinical outcomes of PPFOC have been recently investigated in 388 patients older than 60 years who had undergone a PPFOC after a potential PFO-related ischemic accident. Lower rates of recurrent stroke/TIA were reported than was expected in a population of elderly patients receiving PPFOC for CS [72]. Moreover, an unknown AF incidence in the post-procedural period was more common in older patients [72]. Nevertheless, early outcomes for elderly people remain controversial [73,74,75,76]. Although these findings suggest that PPFOC and APT might be considered in the elderly [71,72], further studies would be needed to confirm this hypothesis.

Furthermore, studies comparing medical therapy with PPFOC in patients >60 years of age, especially without coronary risk factors, are ongoing [63,77]. In addition, the analysis of actual randomized controlled trials supports PPFOC with multiple device types, although the US Food and Drug Administration had approved Amplatzer and Gore Septal Occluder devices [78,79,80]. New devices, such as the suture-mediated NobleStitch EL, seem to be promising in terms of efficacy and device-related complications reduction, despite the high costs [81].

## 6. Assessment of Hidden Atrial Fibrillation and Rhythm Evaluation

An AF incidence of 4.9–9.2% has been reported in patients with CS [82,83,84]. However, due to the improvement in AF detection tools, an arrhythmic burden has been recognized in 25% of patients with PFO who developed CS [85,86,87,88]. Long-term cardiac monitoring (CM) with continuous wearable recorders or implantable loop recorders (ILR) has increased the likelihood of detecting subclinical AF [89,90,91]. It is worth mentioning that in patients with CS and PFO, AF should be excluded before considering the interventional approach [92]. Indeed, if AF is known, the routine PPFOC should not be generally recommended according to the latest international guidelines [92]; this is because in the presence of AF, PFO might be “an innocent bystander” even though it has high-risk features. In this case, PPFOC and APT could represent a suboptimal approach implying stroke recurrences, whereas OAC might be the most appropriate choice. Therefore, detecting AF plays a crucial role in the decision-making of PFO patients. However, succeeding in AF recognition through traditional short-term monitoring may be particularly challenging, since it often occurs in an asymptomatic form [93]. Nevertheless, several consensus documents and guidelines have addressed the evidence about long-term CM and the optimal monitoring tools for assessing the presence of AF [15,16,94,95,96]. According to a SCAI expert consensus statement [94], the duration of CM should last approximately 4 weeks in patients ≤ 40 years and for 1–2 weeks in those >40 years, providing that other AF risk factors such as hypertension, diabetes, hyperthyroidism, valvular heart disease (VHD) or alcohol use do not coexist. Conversely, as stated by Prestipino et al. [14], it could be reasonable to use implantable cardiac monitoring (ICM) lasting more than 6 months before considering PPFOC in patients aged >65, in those aged 55–64 years at risk for AF, and in those aged <55 years with more than two AF high-risk factors if routine negative monitoring (12-lead ECG, cardiac telemetry, and 24-h Holter) had not shown AF. The ESO-Guidelines [95] suggested that subjects with cerebral ischemic events of undetermined origin prolong CM for more than 24 h using implantable devices. According to the American Heart Association (AHA)/American Stroke Association (ASA) Guidelines, in the absence of OAC contraindications, a long-term CM with telemetry or IRL for detecting intermittent AF is considered a reasonable approach [15]. Currently, several non-invasive ECG monitoring tools (such as ECG, telemetry, inpatient or outpatient Holter monitoring, external ambulatory ECG recorders, and continuous wearable ECG recorders) and ICM have been proposed for a long-term CM to improve AF detection [91]. ICM and 24/48 h ECG Holter have been shown to be performed in 80% and 30% of CS patients by a cross-sectional European survey [97] from 79 centers in 34 countries, reporting them as the most common monitoring instruments used for investigating AF in the CS field. The post-stroke AF identification by CM methods has been investigated in a meta-analysis [98] which divided the post-stroke period into four phases: (1) emergency room period; (2) in hospital period; (3) first ambulatory period; (4) second ambulatory period. Admission ECG (phase 1), serial ECG, continuous inpatient ECG monitoring or telemetry and in-hospital Holter monitoring (phase 2), ambulatory Holter (phase 3), mobile cardiac outpatient telemetry, external loop recording, and IRL (phase 4) revealed an AF detection rate of 7.7%, 5.1%, 10.7%, and 16.9%, respectively. Finally, AF has been found in 23.7% of overall phases [98]. In the Embrace study, conducted on 572 subjects aged ≥ 55 years who developed CS in the absence of AF history within the previous 6 months [99], the ambulatory ECG monitoring for 30 days has been shown to improve AF diagnosis compared with standard 24–48 Holter monitoring, with an AF detection rate of 16.1% vs. 3.2%. Accordingly, in CRYSTAL AF, which enrolled 441 adults with CS, ICM long-term monitoring has been demonstrated to be more effective than traditional follow-up. Indeed, AF has been detected in 8.9% and 1.4% within 6 months of using ICM and conventional follow-up, respectively [100]. Moreover, the superiority of ICM has been confirmed at 12 months, revealing 12.4% of AF in the interventional group vs. 2.0% in the non-interventional group [100]. Regarding the duration of CM, the time for detecting AF optimal remains controversial. However, prolonging the monitoring period over 24 h has been associated with improved detection of AF. In a meta-analysis of 8215 patients with CS, an increase in the duration of ICM follow-up resulted in an improvement of AF diagnosis from 2.0% to 28.5% at 1 week control and at 36 FU, respectively [101]. ILR monitoring has been shown to be superior to external loop recorders in a recent trial PERDIEM, on 300 randomized patients with Cs. Indeed, AF diagnosis has been reported in 15.3% and 4.7% of patients who received ILR and external recorders lasting 30 days, respectively [102]. These results have been confirmed by the LOOP study, which found a three-fold increase in AF detection in patients who underwent the ILR strategy [89]. The FIND-AF trial conducted on subjects with ischemic stroke in the absence of AF history, including CS, established that 10 days of ECG-Holter monitoring was more effective than 24 h ECG-Holter at baseline at 3 and 6 months follow-up, with an AF detection rate of 14% and 5%, respectively [103]. In accordance, the STROKE-AF [104] trial confirmed the superiority of ILR over traditional CM. ECG monitoring with an ICM was superior to conventional follow-up for detecting AF after CS.

Recognizing that AF is important not only before but also after PPFOC is essential. Indeed, AF frequently occurs after PPFOC with an incidence of 4.6–6.6% [13,105], mostly complicating the periprocedural period. A five-fold increased risk of AF has been estimated in patients who undergo the procedure, especially by the first 45 days, although they remain at risk for longer [106]. In the early post-procedure period, the mechanisms for initiating AF may be related to both anatomical–physiological and device–procedure determinants. In particular, device-related mechanical manipulation may induce an inflammatory response [107]. Nevertheless, whether procedural-related inflammation triggers the development of AF, or whether an unrecognized pre-existing arrhythmic burden exists, remains unclear [60]. However, it is worth mentioning that subclinical events are generally not detected, so that AF prevalence may be largely underrecognized. In a study conducted on 225 patients without known AF who had undergone PPFOC within 1 month after the procedure, monitored by internal or external loop recorders in the follow-up, supraventricular tachycardias (SVT) ≥ 30 s and AF have been reported in 20.9% and 80%, respectively [108]. What should be highlighted the most is that recurrent arrhythmic episodes were recorded in 40% of patients, 11.8% of whom had arrhythmias lasting ≥ 24 h. Nevertheless, a longer-term CM in patients with PFOs and CS should be considered [108], although under which circumstances this approach may be advisable remains debated. Moreover, it has been hypothesized that some devices may be more likely associated with AF. Indeed, the Amplatzer PFO Occluder has been thought to have a lower rate of incident AF after PPFOC than the GORE Occluder [107]. However, these findings must be more adequately confirmed. The antithrombotic strategy in patients with postprocedural AF remains debated. Several criteria have been adopted to lead the clinical decision of long-term antithrombotic therapy, such as the recurrence of AF episodes following a successful cardioversion [109,110,111,112], AF episodes lasting more than 48 h [109,112], and the occurrence of AF after the early postprocedural phase [74,110]. A short-term (<6 months) antithrombotic period might represent an agreeable choice [74]. However, the OAC approach should be tailored, and several factors, such as CHA2DS2 VASc score and bleeding risk, should be considered.

## 7. Antiplatelet Strategy after PFO Closure

Dual antiplatelet therapy (DAPT) is generally adopted for 1–6 months after PPFOC [14,50]. However, another approach may consist of continuing DAPT for only 6–8 weeks. After the DAPT period, it is considered reasonable to continue with a single antiplatelet agent (usually aspirin) for 1–5 years [14,50]. On the other hand, lifelong APT use may also be considered [97,113]. Notably, the preferred approach has not been well established due to the lack of evidence-based data. On the contrary, patients who underwent long-term APT are likely to be associated with an increased risk of major bleeding [114,115,116]. Moreover, the bleeding risk seems to outweigh the ischemic one [113]. However, considering the fact that ad hoc trials are missing, there is no total agreement on this issue, and the strategy may depend on the patient’s characteristics and comorbidities. Nevertheless, it is necessary to take into account several aspects. Firstly, a device-related thrombus is a potential complication [117,118,119] that occurs in 1–2% of patients [95,120]. Its incidence is higher by the first month after PPFOC, whereas it diminishes after 1 year [9]. Secondly, it is important to consider that endothelial modifications physiologically follow PPFOC [121]. A complete device endothelialization is expected to be achieved within 6 months after PPFOC [122]. However, it has been proposed that the endothelialization process of the device may last for more than 5 years after the procedure [122]. Notably, this process is associated with the activation of cascade coagulation [123,124] and may imply harmful complications such as stroke and systemic embolism [120]. Consequently, prolonging a DAPT for 6 months and then continuing for a long-term period to prevent potential device thrombosis ischemic-related events may be advisable. Conversely, a lifelong single APT approach might not be necessary if PPFOC has been effective, in the absence of significant residual shunt and co-morbidities. Moreover, the DAPT discontinuation strategy at 6 months compared to an indefinite assumption has been shown to not be different in terms of ischemic stroke, but to increase bleeding [125,126,127]. However, no data exist on the cessation of APT after PPFOC.

## 8. Conclusions

New meta-analyses and RCTs have helped us to identify high-risk patients who may benefit from PFO closure to learn about subsequent complications such as AF.

Considering the data currently available, in selected patients, with a moderate-to-large shunt as a part of a comprehensive clinical evaluation, it seems reasonable to assume that the PFO closure is an option superior to medical therapy for the secondary prevention in patients with previous CS.

Collaboration with neurologists is indispensable, and future studies will better guide the closure of PFO in CS, together with the introduction of new devices.

## Figures and Tables

**Figure 1 jcm-12-01952-f001:**
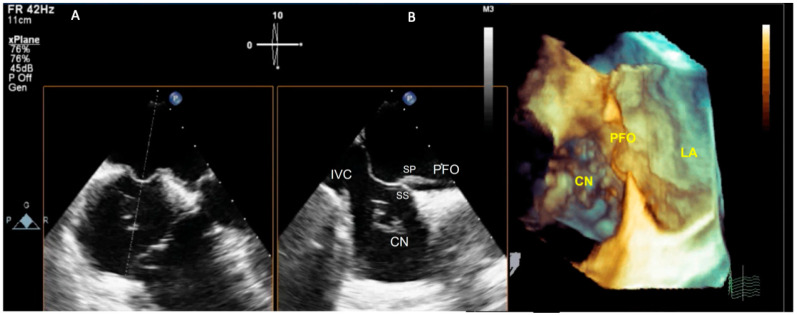
(**A**) Mid-esophageal four-chamber view at 0 degrees and mid-esophageal bi-caval view at 90 degrees, simultaneously obtained by 2D-TEE X-plane modality. Simultaneous biplane mode (X-plane) allows us to obtain two simultaneous 2D images. (**B**) Cropped 3D-TEE image with zoom 3D modality. A complex tunnel PFO is shown. The 3D-TEE image shows a long septum primum (SP), widely separated from the septum secundum (SS). The Chiari network (CN) in the right atrium is displayed. IVC: inferior vena cava; SP: septum primum; PFO: patent formen ovale; SS: septum secundum; CN: Chiari network; LA: left atrium.

**Figure 2 jcm-12-01952-f002:**
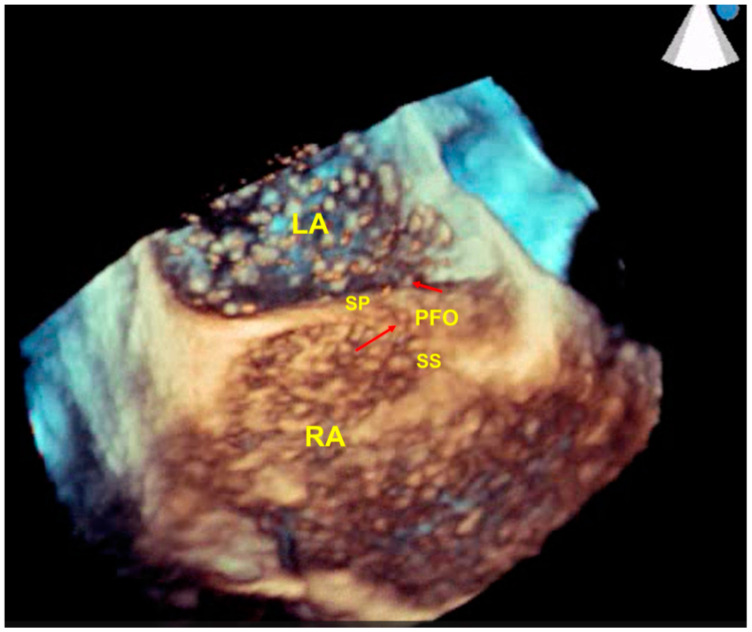
A 3D-TEE real-time microbubble test. Microbubbles opacify the right atrium (RA). Several microbubbles (red arrows) pass through the PFO, reaching the left atrium (LA) from the RA. SP: septum primum; PFO: patent formen ovale; SS: septum secundum; RA: right atrium.

**Table 1 jcm-12-01952-t001:** RoPE score calculator (adapted by Kent et al., 2013).

RoPE Score Calculator	
Absence of hypertension	+1
Absence of diabetes	+1
Absence of stroke/TIA	+1
Nonsmoker	+1
Cortical infarct, on imaging	+1
Age	
18–29	+5
30–39	+4
40–49	+3
50–59	+2
60–69	+1
>70	0

**Table 2 jcm-12-01952-t002:** Pascal classification (adapted by Kent 2020).

Risk Source	Features
Very high	A PFO and a straddling thrombus
High	Concomitant pulmonary embolism or deep venous thrombosis preceding an index infarct, combined with either a PFO and ASA or a large-shunt PFO
Medium	A PFO and an ASA or a large-shunt PFO
Low	A small-shunt PFO without an atrial septal aneurysm

## Data Availability

Not applicable.

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
