# Peer review of "Patent Foramen Ovale and Cryptogenic Stroke: Integrated Management"

_jcm, 2023, doi:10.3390/jcm12051952_

Round 1

Reviewer 1 Report

This is a nice summary of the current status of PFO closure. I do not think it adds much additional information to the guideline documents which have been recently published. I would add Jonathan Tobis's manuscript on the Proposal for updated nomenclature for classification of PFA-associated stroke. JAMA neurol 2020 77(7):878.

Author Response

REVIEWER 1

This is a nice summary of the current status of PFO closure. I do not think it adds much additional information to the guideline documents which have been recently published. I would add Jonathan Tobis's manuscript on the Proposal for updated nomenclature for classification of PFA-associated stroke. JAMA neurol 2020 77(7):878.

ANSWER: We agree with the reviewer. We changed accordingly. We included the citation of  Jonathan Tobis's manuscript on the Proposal for updated nomenclature for the classification of PFA-associated stroke. JAMA neurol 2020 77(7):878. See page 2, lines 70-78

Reviewer 2 Report

Pinto PG, et al tried to present a review for PFO management in this paper. This paper is informative to stratify the treatment of PFO, but there are several issues to be addressed.

1. It is importatnt to describe the assessment of hidden atrial fibrillation, escpcially recommendation of rhythm evaluation.

2. Anit-platelet strategy is also interesting after PFO closure. It would be better to add this part.

Author Response

REVIEWER 2

Pinto PG, et al tried to present a review for PFO management in this paper. This paper is informative to stratify the treatment of PFO, but there are several issues to be addressed.

ANSWER: We thank the reviewer for his/her kind comments. We changed accordingly.

  1. It is importatnt to describe the assessment of hidden atrial fibrillation, especially recommendation of rhythm evaluation. ANSWER:We agree with the reviewer. We added a section on the assessment of hidden atrial fibrillation and rhythm evaluation before and after PFO closure. See pages 6-7, lines 243-341
  2. Anit-platelet strategy is also interesting after PFO closure. It would be better to add this part. ANSWER:We agree with the reviewer about the importance of the antiplatelet strategy after PFO closure. We added this part. See pages 8, lines 342-368

Reviewer 3 Report

This is a well-written and nice review about the current status of PFO closure for patients with cryptogenic stroke. This reviewer has just minor comments to add to the current state of the manuscript.

Minor comments:

Line 79: "and" may be deleted, as well as "others"

The GORE-REDUCE trial has the same reference as the CLOSE trial. Please correct.

The following sentence it´s hard to understand and grammatically incorrect: “In a recent meta-analysis, according to the Rope score (Table 1) integrated with the PASCAL classification [60], which describes the anatomical or functional characteristics of the PFO allows us to indicate PFO closure, it has evaluated the probability that a PFO may cause of a cryptogenic stroke” . Please rephrase it.

When discussing about the convenience of performing PFO closure for patients above 60 years of age with cryptogenic stroke, a recent manuscript has demonstrated lower rates of recurrent stroke/TIA than expected in a population of older patients receiving PFO closure for CS. (Transcatheter Closure of Patent Foramen Ovale in Older Patients with Cryptogenic Thromboembolic Events. Circ Cardiovasc Interv. 2022;15(7):557-565. doi:10.1161/CIRCINTERVENTIONS.121.011652). It would be interesting to add this evidence to the manuscript.

Author Response

REVIEWER 3

This is a well-written and nice review about the current status of PFO closure for patients with cryptogenic stroke. This reviewer has just minor comments to add to the current state of the manuscript.

ANSWER:We thank the reviewer for his/her kind comments. We changed accordingly.

Minor comments:

  1. Line 79: "and" may be deleted, as well as "others" . ANSWER:We corrected.
  2. The GORE-REDUCE trial has the same reference as the CLOSE trial. Please correct. ANSWER:We corrected.
  3. The following sentence it´s hard to understand and grammatically incorrect: “In a recent meta-analysis, according to the Rope score (Table 1) integrated with the PASCAL classification [60], which describes the anatomical or functional characteristics of the PFO allows us to indicate PFO closure, it has evaluated the probability that a PFO may cause of a cryptogenic stroke” . Please rephrase it.

ANSWER:We agree with the reviewer.  We rephrased and explained this point more extensively. See page 5, lines 203-213

When discussing about the convenience of performing PFO closure for patients above 60 years of age with cryptogenic stroke, a recent manuscript has demonstrated lower rates of recurrent stroke/TIA than expected in a population of older patients receiving PFO closure for CS. (Transcatheter Closure of Patent Foramen Ovale in Older Patients with Cryptogenic Thromboembolic Events. Circ Cardiovasc Interv. 2022;15(7):557-565. doi:10.1161/CIRCINTERVENTIONS.121.011652). It would be interesting to add this evidence to the manuscript. ANSWER:We agree with the reviewer.  We have added and discussed this reference. Se page 6, lines 228-236

Round 2

Reviewer 1 Report

manuscript acceptable

Reviewer 2 Report

Authors responded appropriately and added descriptions that I suggested. I have no more comment.